# Single-cell RNA-seq reveals heterogeneity in hiPSC-derived muscle progenitors and E2F family as a key regulator of proliferation

Minas Nalbandian[1], Mingming Zhao[1], Hiroki Kato[1,2], Tatsuya Jonouchi[1], May Nakajima-Koyama[3], Takuya Yamamoto[3,4,5], Hidetoshi Sakurai[1]

**Human pluripotent stem cell-derived muscle progenitor cells (hiPSC-MuPCs) resemble fetal-stage muscle progenitor cells and possess in vivo regeneration capacity. However, the heterogeneity of hiPSC-MuPCs is unknown, which could impact the regenerative potential of these cells. Here, we established an hiPSC-MuPC atlas by performing single-cell RNA sequencing of hiPSC-MuPC cultures. Bioinformatic analysis revealed four cell clusters for hiPSC-MuPCs: *myocytes*, *committed*, *cycling*, and *noncycling progenitors*. Using FGFR4 as a marker for *noncycling progenitors* and *cycling* cells and CD36 as a marker for *committed* and *myocyte* cells, we found that FGFR4+ cells possess a higher regenerative capacity than CD36+ cells. We also identified the family of E2F transcription factors are key regulators of hiPSC-MuPC proliferation. Our study provides insights on the purification of hiPSC-MuPCs with higher regenerative potential and increases the understanding of the transcriptional regulation of hiPSC-MuPCs.**

## Introduction

Skeletal muscle satellite cells (i.e., skeletal muscle adult stem cells) confer a high regeneration capacity to the muscle tissue (Mauro, 1961; Relaix & Zammit, 2012). Satellite cells have the capacity to proliferate and differentiate into myoblasts, which fuse to muscle fibers when needed for regeneration (Baghdadi & Tajbakhsh, 2018; Evano & Tajbakhsh, 2018). Owing to this, satellite cells have been studied in cell therapies for skeletal muscle disease such as Duchenne muscular dystrophy (DMD) (Montarras et al, 2005; Cerletti et al, 2008; Sacco et al, 2008; Tanaka et al, 2009; Marg et al, 2014; Xu et al, 2015; Nalbandian et al, 2021). However, because they are difficult to obtain at high numbers and lose their regenerative

potential after in vitro expansion (Montarras et al, 2005; Negroni et al, 2009; Gilbert et al, 2010), alternative methods have been developed to produce cells that resemble satellite cells in terms of in vivo myogenic potential.

As one example, several groups have successfully developed methods for the myogenic induction of human induced pluripotent stems (hiPSCs) (Darabi et al, 2012; Tanaka et al, 2013; Shelton et al, 2014; Chal et al, 2015; Xi et al, 2017; Wu et al, 2018; Sato et al, 2019; Zhao et al, 2020). The resulting muscle cells serve as in vitro models to study development (Lilja et al, 2017; Magli & Perlingeiro, 2017; Al Tanoury et al, 2020; Xi et al, 2020) and disease modeling (Sun et al, 2020; Al Tanoury et al, 2021; Uchimura et al, 2021). Moreover, hiPSC-derived myogenic cell cultures not only include myotubes but also muscle progenitors (hiPSC-MuPCs) (Shelton et al, 2014; Chal et al, 2015; Xi et al, 2020; Zhao et al, 2020). HiPSC-MuPCs have been used as alternatives to satellite cells in the study of cell therapies for DMD. HiPSC-MuPCs more resemble fetal MuPCs rather than mature satellite cells (Incitti et al, 2019; Xi et al, 2020; Zhao et al, 2020; Nalbandian et al, 2021). Notably, fetal MuPCs and adult skeletal muscle stem cells have different transcriptomes (Xi et al, 2020) and functions (Tierney et al, 2016). Furthermore, fetal MuPCs are characterized by a relative high number of cells (Xi et al, 2020) and are in a proliferative state, which is believed to be critical for the formation of skeletal muscle during developmental stages.

Despite the above studies, little is known about the cell heterogeneity of hiPSC-MuPCs. Such knowledge would help to identify the best type for cell therapies. In the present research, by performing single-cell RNA sequencing (scRNA-seq) of hiPSC-MuPCs cultures, we studied the cell heterogeneity of the myogenic subset of cells, finding four clusters of cells: *noncycling progenitors*, *cycling*, *committed*, and *myocytes*. Furthermore, using FGFR4 and a newly reported marker, CD36, we could sort two fractions of hiPSC-MuPCs: one more stem cell-like and the other more myocyte cell–like. These cells populations showed differences in the gene expressions of myogenic markers, morphology, and in in vitro and

---

[1]Department of Clinical Application, Center for iPS Cell Research and Application (CiRA), Kyoto University, Kyoto, Japan   [2]Asahi Kasei Co., Ltd., Tokyo, Japan   [3]Department of Life Science Frontiers, Center for iPS Cell Research and Application (CiRA), Kyoto University, Kyoto, Japan   [4]Institute for the Advanced Study of Human Biology (WPI-ASHBi), Kyoto University, Kyoto, Japan   [5]Medical-risk Avoidance Based on iPS Cells Team, RIKEN Center for Advanced Intelligence Project (AIP), Kyoto, Japan

Correspondence: hsakurai@cira.kyoto-u.ac.jp; mnalband@stanford.edu

in vivo myogenic capacity, which indicated the stem cell–like MuPCs are most suitable for cell transplantation. Furthermore, by analyzing the single-cell transcriptome, we described the transcription factor (TF) gene expression landscape across cell populations and identified the E2F family as key players of the cell proliferation.

## Results

### hiPSC-MuPCs are a heterogeneous cell population

To study hiPSC-MuPCs, we differentiated hiPSCs to the myogenic lineage using a stepwise differentiation protocol (Fig 1A) (Zhao et al, 2020). After 80 d of culture, we performed a histochemical analysis and found several hiPSC-MuPCs that expressed PAX7 and/or MYOD1 and surrounded myotubes expressing myosin heavy chain (Fig 1B). To quantify the hiPSC-MuPCs, we dissociated them into single cells and seeded the mononuclear cells for immunocytochemistry. Quantification of stained mononuclear cells revealed cell heterogeneity in the hiPSC-MuPC population (Fig 1C), with three types of populations: PAX7+/MYOD1−, PAX7+/MYOD1+, and PAX7−/MYOD1+. These results prompted us to further study cell heterogeneity. For this purpose, we generated a transcriptomic atlas of hiPSC-MuPCs cultures differentiated for 80 d by performing single-cell RNA sequencing (scRNA-seq).

To process the scRNA-seq data, we used the Seurat package (Butler et al, 2018). Cells of two different batches were analyzed separately. After filtering conditions were applied, we compared datasets from the different batches, finding no significant differences between them (Fig S1A). To increase the sample size, we combined the two batches for subsequent analysis, resulting in 5,318 total number of cells. Samples were assembled into a cell atlas using a uniform main-fold approximation and projection (UMAP) to observe gene expressions. A clustering analysis revealed three main clusters and one small cluster. The small cluster (named *unknown cells*) was composed of only a few cells and not considered for subsequent analysis. We interpreted the main clusters as three different cell populations, and based on the normalized gene expression of classical cell markers, we defined them as "*myogenic population*," "*mesenchymal cells*," and "*neuronal cells*" (Figs 1D and S1B).

*Myogenic population* cells expressed the classical MuPC markers *PAX7, MYF5, MYOD1*, and *MYOG*. *Mesenchymal cells* expressed *PDGFRA* (Uezumi et al, 2014), *PDGFRB, ENG, COL6A2*, and *COL6A3*, and *neuronal cells* expressed the neuronal progenitor cells marker *SOX1, SOX2, SOX3, SOX6*, and *PAX6* (Ellis et al, 2004) (Figs 1E and F and S1C). A Gene Ontology (GO) analysis for biological process of the differentially expressed genes (DEGs) in each group was performed. DEGs up-regulated in the *muscle population* were enriched for terms related to muscle formation and skeletal muscle development; DEGs up-regulated in *mesenchymal cells* were enriched for terms like *extracellular matrix organization* and *collagen fibril organization*; and DEGs up-regulated in *neuronal cells* were enriched for terms related to neuron development (Fig 1G).

To gain deeper insights in the cell heterogeneity of hiPSC-MuPCs, we then performed a clustering analysis, finding four subpopulations (Fig 2A). We defined the "*noncycling progenitors*" population

to include cells expressing *PAX7* and *MYF5* but not *MKI67*; the "*cycling*" population to include cells expressing *PAX7, MYF5* and *MKI67*; the "committed" population to include cells expressing *MYOG* and *MYOD1*, but not *PAX7, MYF5*, or *MKI67*; and the "myocytes" population to include cells expressing *MEF2C, MYH3, MYOG*, and *MYOD1* but not *PAX7, MYF5*, or *MKI67* (Figs 2A–C and S2A). We then used the scRNA-seq data to analyze the cell cycle status of hiPSC-MuPCs (Kowalczyk et al, 2015) and confirmed that most *cycling* cells were transitioning from G2/M and to S, whereas *noncycling progenitors, myocytes*, and *committed* cells had exited the cell cycle (Fig 2D). Analysis of the gene expression of cell cycling–related genes confirmed a higher expression by *cycling cells* (Fig S2B). Finally, we performed a GO analysis of the DEGs for each group (Figs 2E and S2C). Consistent with the cell types, *cycling* cells were enriched for terms related to the cell cycle; *committed* cells were enriched for terms related to muscle differentiation, and *myocytes* were enriched for terms related to myofiber assembly and muscle contraction. On the other hand, *noncycling progenitor* cells were enriched for terms related with protein synthesis and RNA catabolic process, suggesting increased protein synthesis. Furthermore, an analysis of DEGs between *noncycling progenitors* and *cycling progenitors* revealed that almost all the DEGs were genes up-regulated in the *cycling progenitors* (Fig S2D), indicating that the cycling progenitors activated several stage-specific genes. GO analysis confirmed that the up-regulated genes in the *cycling progenitors* enriched for cell proliferation–related terms (Table S1).

Because TFs likely play a major role in each myogenic subpopulation, we searched for enriched TFs binding site motifs in the promoter regions of the up-regulated DEGs in each myogenic cluster. This search revealed several gene family recognition motifs for each group (Fig 3A). We also analyzed the expression of TFs that bind to the most enriched binding motifs of each cell cluster. We could not identify TFs that were uniquely expressed in the *noncycling progenitors* population for the binding sites motifs of the top up-regulated DEGs. *E2F1, E2F2*, and *E2F7* were found to be exclusively expressed by *cycling* cells, which also showed enrichment for their binding motifs in the promoter regions of up-regulated DEGs (Fig S3A and B). Furthermore, the predicted downstream target genes of the E2F family in *cycling* cells were enriched for cell cycle and proliferation related terms (Table S2), strengthening the possibility that E2F family genes play a major role in proliferation, as previously reported (Yan et al, 2003). In a similar fashion, *MYOD1* expression was enriched in *committed* cells (Fig 2C), and the predicted downstream target genes were enriched for skeletal muscle tissue development-related terms (Table S2), suggesting a major role of MYOD1 in the transcriptomic control of *committed* cells. Moreover, we identified three different binding sites in *committed* cells for the TF *SRF*, which was highly expressed by *committed* cells (Fig S3C). A GO analysis of the DEGs up-regulated in *committed* cells with binding sites for SRF in their promoter region showed an enrichment for terms related with muscle formation (Table S2), suggesting a role in myogenic differentiation (Randrianarison-Huetz et al, 2018). In the *myocytes* population, *MEF2C* was highly expressed (Fig 2C). Predicted MEF2C downstream target genes were enriched for GO terms related to muscle contraction and myofibers structure (Table S2), suggesting an important role of MEF2C in the late stages of myogenesis, as previously reported (Dodou et al, 2003; Piasecka

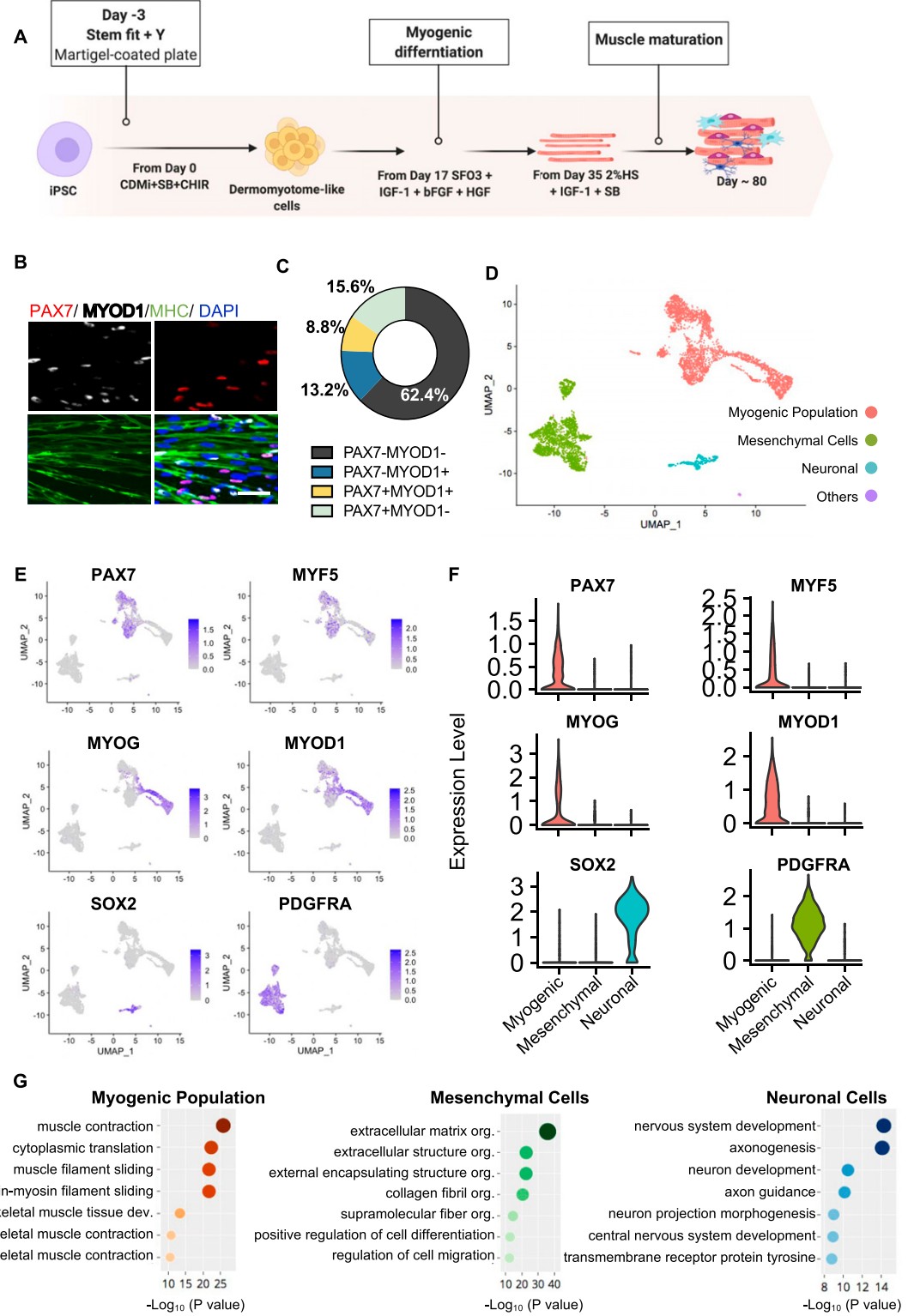

**Figure 1. Transcriptomic Atlas of hiPSC-derived muscle progenitor cells cultures.**
**(A)** Schematic representation of the myogenic induction protocol. Cells were analyzed after 80 d of differentiation. CHIR, CHIR99021; HS, horse serum; SB, SB431542.
**(B)** Immunohistochemical analysis of PAX7 (green), MYOD1 (white), MYH (MYOSIN HEAVY CHAIN, green), and DAPI (blue) at day 80 of the differentiation. Scale bar, 200 μm.
**(C)** Quantification of PAX7 and MYOD in dissociated hiPSC-MuPCs at day 80 of the differentiation. Data represent the mean of three independent experiments.
**(B, D)** A transcriptomic atlas of all cells dissociated from the cells in (B). **(E)** Single-cell expression of the selected markers. **(F)** Violin plots showing expression clusters of the selected markers. **(G)** A Gene Ontology analysis (biological processes) for differentially expressed genes up-regulated in each cluster.

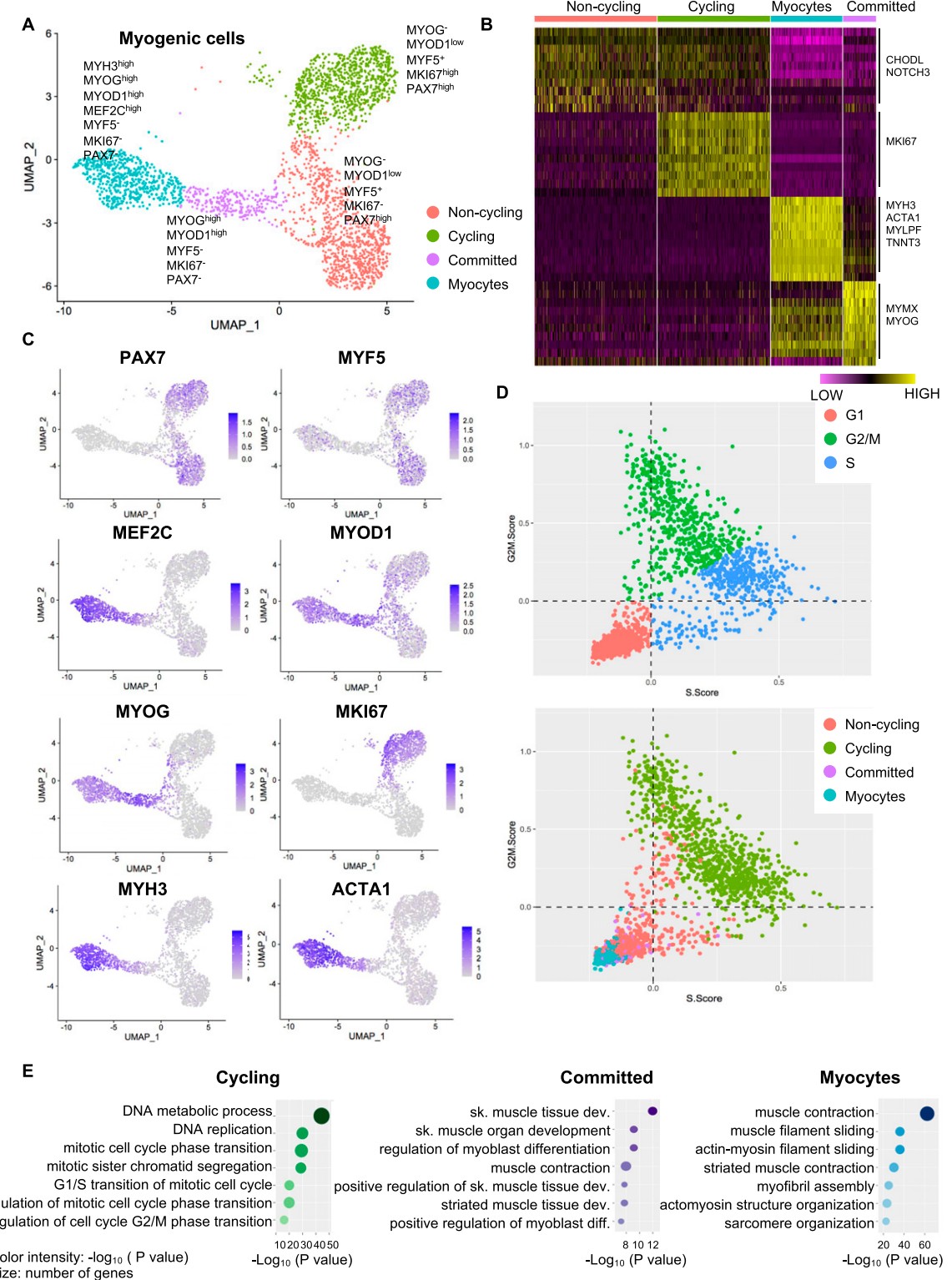

**Figure 2. Heterogeneity among hiPSC-derived muscle progenitor cells.**
**(A)** Transcriptomic atlas and clustering of the subset of myogenic cells in the hiPSC-MuPC cultures. **(B)** Heatmap representing the expression of the differentially expressed genes for each cluster. **(C)** Single-cell expression of selected markers. **(D)** Cell-cycle analysis at the single-cell level. The top panel indicates cell cycle stages, and the lower panel indicates the corresponding clusters. **(E)** A Gene Ontology analysis (biological processes) for differentially expressed genes up-regulated in *cycling* cells, *committed* cells, and *myocytes*.

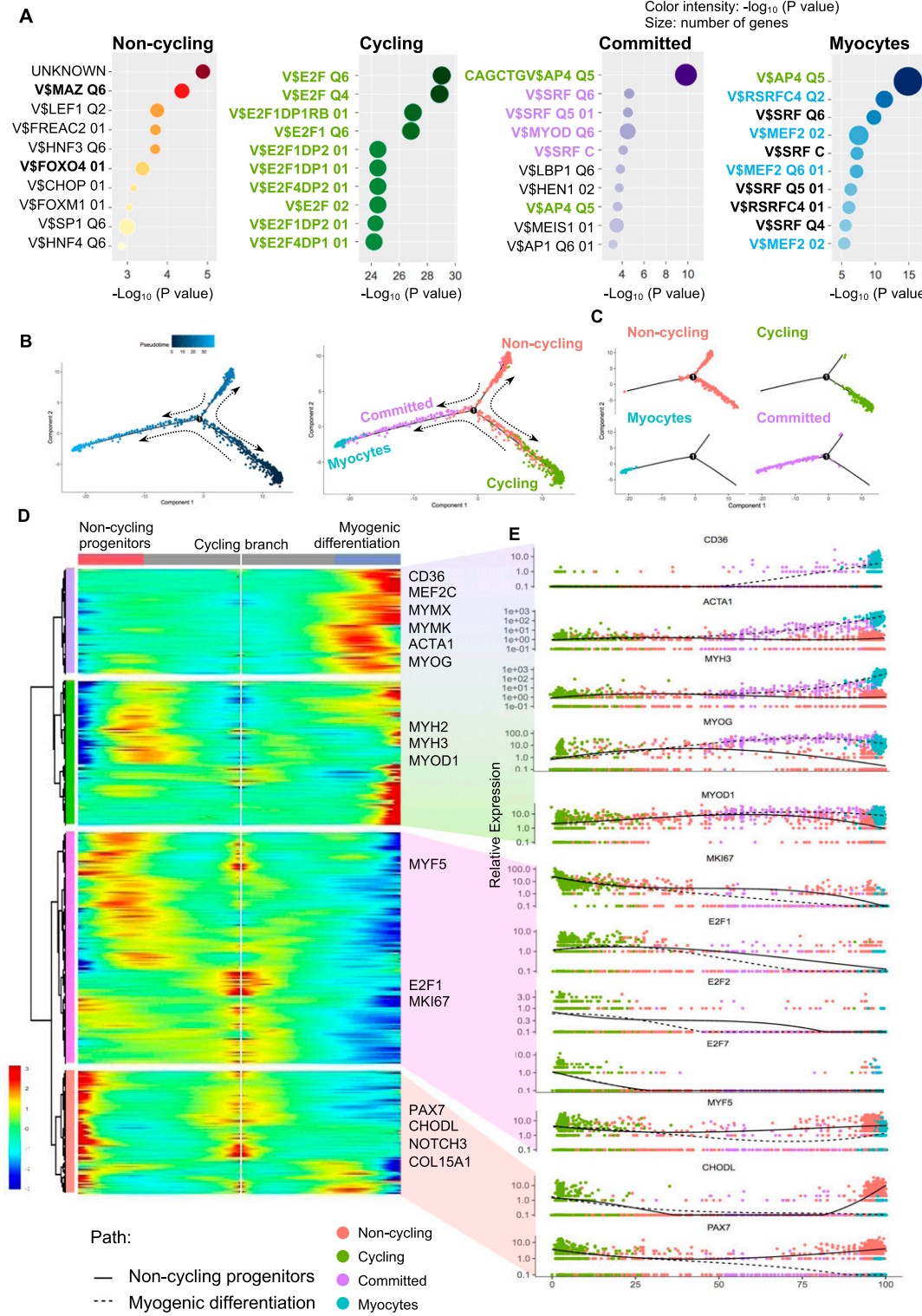

**Figure 3. Transcriptional regulation of different population of hiPSC-derived muscle progenitor cells and pseudotime analysis.**
**(A)** Top enriched TF-binding site motifs in the promoter region of the differentially expressed genes for each cluster. Binding sites in bold green are bound by TFs expressed in *cycling* cells, bold purple are bound by TFs expressed in *committed* cells, bold blue are bound by TFs expressed in *myocytes*, and bold black represent are bound by TFs expressed in all cell clusters. **(B)** Pseudotime analysis performed with the Monocle package. The left plot shows the cell hierarchy in the pseudotime trajectory, and the right plot shows the location of the different clusters in the pseudotime plot. **(C)** Location of cells in the pseudotime trajectory branches. **(D)** A heatmap showing the expression of the differentially expressed genes for each branch in the pseudotime analysis. Each column represents a cell, and each row represents a gene. **(D, E)** Pseudotime ordered single-cell gene expression of representative genes selected from (D).

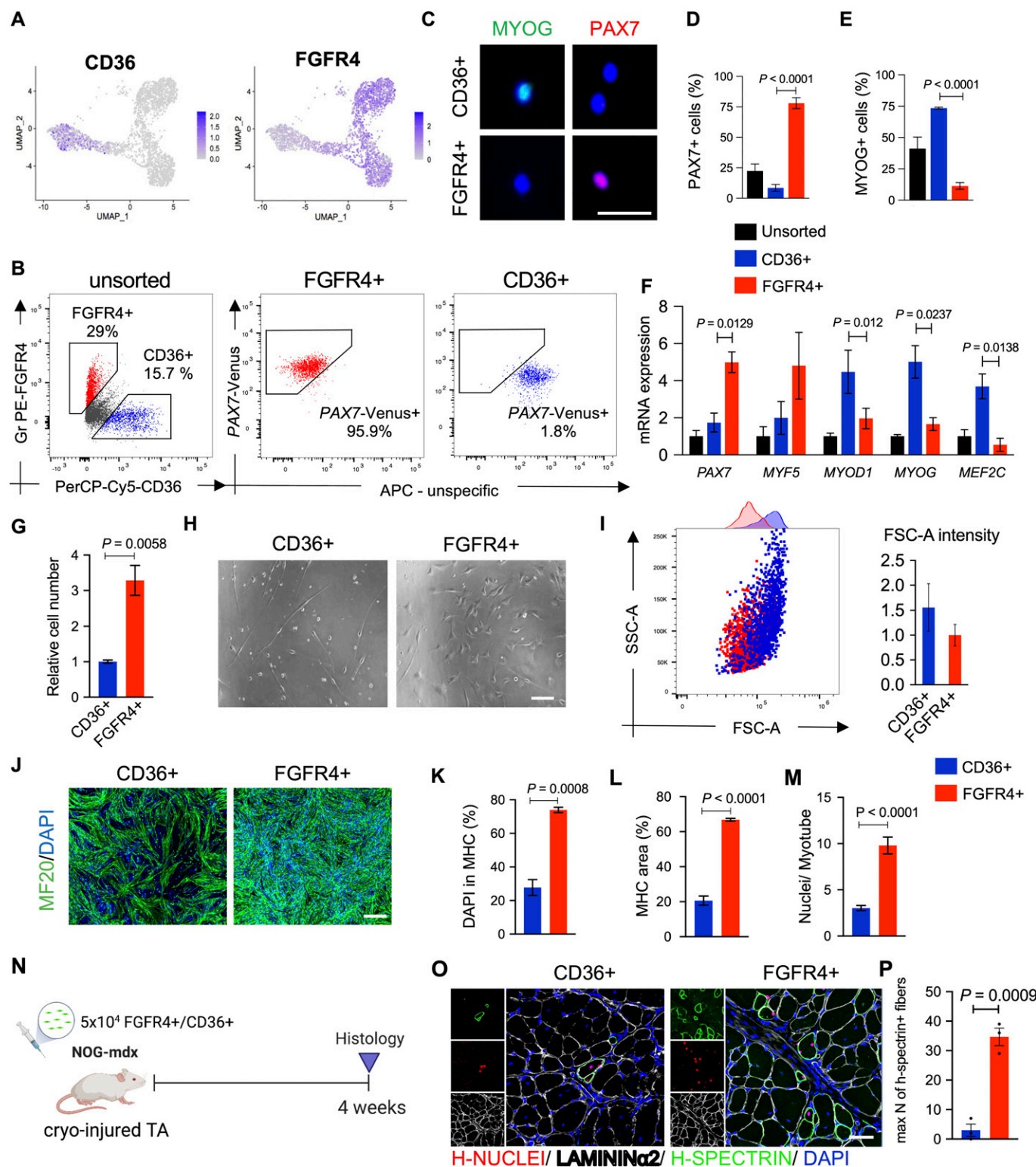

**Figure 4. Myogenic capacity of FGFR4+ and CD36⁺ hiPSC-derived muscle progenitor cells.**
**(A)** Gene expression at the single-cell level of FGFR4 and CD36. **(B)** Plots show representative FACS analysis of hiPSC-MuPCs for FGFR4 (PE-conjugated antibody) and CD36 (PerCP-Cy5–conjugated antibody) of MuPCs derived from a *PAX7*-Venus reporter hiPSC line and pre-gated with FGFR4 or CD36. **(C)** Representative histochemical analysis of hiPSC-MuPCs sorted with FGFR4 or CD36 antibodies. PAX7 (red), MYOG (green), and DAPI (blue). Scale bar, 25 $\mu$m. **(D)** Quantification of PAX7+ cells from hiPSC-MuPCs sorted with FGFR4 or CD36. **(E)** Quantification of MYOD+ cells from hiPSC-MuPCs sorted with FGFR4 or CD36. **(F)** Gene expression of myogenic markers in hiPSC-MuPCs sorted with FGFR4 or CD36 antibodies. Gene expression was normalized to *GAPDH*. 414C2 hiPSCs were used. **(G)** Quantification of the relative number of hiPSC-MuPCs sorted with FGFR4 or CD36 antibodies and cultured for 5 d. 414C2 hiPSCs were used. **(H)** Representative images of hiPSC-MuPCs sorted with FGFR4 or CD36 and cultured for 3 d. Scale bar,

et al, 2021). Interestingly the TFAP4-binding site motif (V$AP4Q5) was enriched in the promoter regions of DEGs up-regulated in *myocytes* and *committed* cells, but *TFAP4* itself was enriched in *noncycling progenitors* and *cycling* cells (Fig S3C), suggesting that TFAP4 initiates the transition from *noncycling progenitors* and *cycling* to *committed* and *myocytes* cells. Predicted downstream target genes of TFAP4 were enriched for terms related to muscle contraction and muscle development in *committed* and *myocytes* cells (Table S2). The transcriptomic landscape is graphically summarized in Fig S3D.

To compare these results with the developmental scenario, we evaluated an RNA sequencing database of primary fetal MuPCs and cultured fetal MuPCs (GEO: GSE87365). We found that DEGs up-regulated in the primary cells were enriched for GO terms related to the cell cycle (Fig S4A). When searching for enriched TF-binding motifs from the promoter regions of the DEGs up-regulated in the primary cells, we found E2F as the most enriched TF-binding motif, similar to *cycling* cells (Fig S4B). On the other hand, among the most enriched binding motif promoter regions of up-regulated DEGs in cultured fetal MuPCs, we found MEF2, consistent with *myocytes* cells (Fig S4C). This analysis suggested a transcriptomic correlation between hiPSC-MuPC populations and primary and cultured fetal MuPCs.

## Hierarchical analysis of hiPSC-MuPCs reveals a trajectory for myogenic commitment

In adult skeletal muscle, the consensus model assumes that quiescent cells become activated, proliferate, and start myogenic differentiation. The developmental scenario is quite different, where, depending on the developmental stage, the cell hetero-geneity and cell fate varies (Xi et al, 2020). To gain deeper insight into the cell fate, we decided to hierarchically order hiPSC-MuPCs by organizing them using a trajectory inference model by applying the Monocle2 package (Qiu et al, 2017). This package allows us to order cells along trajectories that may be interpreted as different cell fates. The model revealed three main branches (Fig 3B and C). Consistent with the cell clustering, *noncycling progenitors* cells were located in one branch, *cycling* cells in another, and the third branch included *committed* and *myocytes* cells, suggesting that *noncycling progenitors* and *cycling* cells can be differentiated to *committed* cells and then to *myocytes* cells for full myogenic dif-ferentiation. Interestingly, *cycling* cells were positioned at the starting point of the cell trajectory, suggesting the possibility that they may be the progenitors of *noncycling progenitors* cells.

Next, we analyzed the gene expression dynamics along the trajectory and found four gene clusters (Figs 3D and E and S4D). The first two clusters consisted of genes whose expressions were

enriched across the *committed-myocytes* branch and included the myogenic TFs *MYOD1*, *MEF2C*, and *MYOG*, the myocytes fusion-related genes *MYMK* and *MYMX*, and the myotube component-related genes *ACTA1*, *MYH2*, and *MYH3*. The third cluster included genes whose expression increased along the *cycling* branch: *E2F1* and *MKI67*, which are markers for cell proliferation. The fourth cluster consisted of genes whose expressions were increased along the *noncycling progenitors* and *cycling* branches and included satellite cell markers such as *PAX7*, *CHODL*, *NOTCH3*, and *COL15A1*. Overall, the pseudotime clustering and gene expression analysis supports the idea of myogenic progression from *cycling* cells to *noncycling progenitors* cells and from *cycling* and *noncycling progenitors* cells to *committed* cells and finally to *myocytes* cells.

## CD36 and FGFR4 allows separation of hiPSC-MuPCs

To better understand the relevance of hiPSC-MuPC heterogeneity for cell transplantation, we decided to study the myogenic potential of the different hiPS-MuPC populations. Based on our previous study (Nalbandian et al, 2021), we used FGFR4 as a surface marker to perform cell sorting. We confirmed FGFR4 expression by *noncycling progenitors* and *cycling* hiPSC-MuPCs (Figs 4A and S5A). Further-more, to sort the fraction of *committed* and *myocytes* cells, we screened for new surface markers among up-regulated DEGs, identifying CD36 as a strong candidate (Fig 4A). The CD36 surface marker has been reported to be expressed by myoblast and to play a role in myoblast fusion during myogenic differentiation (Park et al, 2012). To study the different cell populations, we decided to use two different hiPSC lines: the DMD-corrected cell line which is a DMD patient-derived hiPS cell line lacking exon 44 and was rescued by knocking-in exon 44 (Li et al, 2015) and the 414C2 cell line (Okita et al, 2011). The 414C2 cell line was also used to establish a *Pax7*-Venus reporter cell line as previously described (Nalbandian et al, 2021).

Flow cytometry analysis revealed around 20% of cells were FGFR4+/CD36− and around 15% were CD36+/FGFR4− (Fig 4B). Moreover, using the *PAX7*-Venus reporter cell line, we confirmed that most of FGFR4+ cells (~90%) were *PAX7*-positive, whereas most of CD36+ cells (~98%) were *PAX7*-negative cells (Fig 4B). This result was supported by histochemical analysis post–cell sorting, where we found ~80% of FGFR4+-sorted cells and ~5% of CD36+-sorted cells were PAX7+ (Fig 4C and D). On the other hand, MYOGENIN staining revealed that ~5% and ~75% of FGFR4+ and CD36+ cells, respectively, were positive for MYOGENIN (Fig 4C and E). Consis-tently, the gene expressions of PAX7 and MYF5 were enriched in FGFR4+ cells, and the gene expressions of MYOD1, MYOG, and MEF2C were enriched in CD36+ cells, which was confirmed in the two cell lines (Figs 4F and S5B). Furthermore, we confirmed by cell cycle

50 µm. **(I)** The left plot shows a representative FACS analysis of the forward scatter area (FSC-A) and side scatter area (SSC-A) of hiPSC-MuPCs gated positive for FGFR4 (red) or CD36 (blue). The right plot shows the quantification of the FSC-A intensity. **(J, K, L, M)** In vitro differentiation of hiPSC-MuPCs sorted with FGFR4 or CD36 antibodies. Representative immunofluorescence of myosin heavy chain (MHC, green) and DAPI (blue) in FGFR4+-sorted and CD36+-sorted cells (J). Scale bar, 200 µm. Quantification of the % of DAPI in the MHC+ area (K). Quantification of the MHC area (L). Quantification of the number of nuclei per myofiber (M). 414C2 hiPSCs were used from three independent experiments. **(N)** Schematic representation of the transplantation of hiPSC-MuPCs sorted with FGFR4 or CD36 antibodies. 414C2 hiPSCs were used. **(O)** Representative histochemical analysis of TA muscle 1 mo after transplanted with FGFR4+ or CD36+ hiPSC-MuPCs. H-NUCLEI (red), H-SPECTRIN (green), LAMININα2 (white), and DAPI (blue). Scale bar, 50 µm. n = 3 mice. **(P)** Quantification of the maximum number of H-SPECTRIN+ fibers per section. Error bars in (D, E, F, G), (K, L), and (O) represent the mean ± SEM of three independent experiments (n = 3).

analysis that cells sorted with the FGFR4 antibody included cycling and noncycling cells (Fig S5C). Functional analysis revealed that with the two cells lines, the proliferation capacity was significantly lower in CD36$^+$ cells (Figs 4G and S5D and E), and the morphology of the two cell populations was different (Fig 4H). To gain quantitative insights into the morphological differences, we examined the cells' forward-scatter area (FSC-A) by flow cytometry and confirmed that CD36$^+$ cells are larger than FGFR4+ cells (Fig 4I).

To evaluate the myogenic capacity of the CD36$^+$ and FGFR4+ cell populations, we performed in vitro differentiation post–cell sorting. Histochemical analysis revealed more robust differentiation by FGFR4+ cells (Figs 4J and S5F). These results were confirmed by the percentage of nuclei in the myosin heavy chain area (Figs 4K and S5G), the total myosin heavy chain area (Figs 4L and S5H), and nuclei per myofiber ratio (Figs 4M and S5I). Finally, we decided to compare the in vivo regenerative capacity of CD36$^+$ and FGFR4+ cells by transplanting 50,000 cells into the previously cryo-injured tibialis anterior (TA) muscle of immunodeficient mdx model mice, NOG-mdx (Fig 4N). Four weeks after the transplantation, histochemical analysis revealed a much higher regeneration capacity by the FGFR4+ cells compared with the CD36$^+$ cells, as indicated by the number of H-SPECTRIN+ fibers (Figs 4O and P and S5J). These results together, indicated that CD36 and FGFR4 could distinguish two populations of hiPS-MuPCs with different morphological and functional characteristics: FGFR4+ cells, which resemble cells with larger regenerative potential, and CD36$^+$ cells, which resemble cells with poor regenerative capacity.

### E2F family regulates proliferation of hiPSC-MuPCs

By analyzing the most enriched TFs binding sites motif in the promoter regions of the DEGs genes, we found several TFs as candidate regulators for each of the hiPS-MuPC clusters identified by the single-cell RNA-seq analysis. The most enriched TF-binding site motifs in *cycling* cells were E2F binding sites (Fig 3A). *E2F1*, *E2F2*, and *E2F7* were found to be exclusively expressed in *cycling* cells (Fig S3A and B). By analyzing previously published scRNA-seq data, we found that *E2F* family genes are exclusively expressed during development by MuPCs but not in postnatal stages (Fig S6A). Noting that above, we found *E2F1*, *E2F2*, and *E2F7* as candidate TF regulators of *cycling* cells (Figs 3A and S3A and B), and we hypothesized that these TFs are important during developmental stages, prompting us to study their function in hiPSC-MuPCs.

To begin, we confirmed that *E2F1*, *E2F2*, and *E2F7* were enriched in FGFR4+ cells (Figs 5 and S6B). Then, we silenced the *E2F1*, *E2F2*, and *E2F7* genes separately by transfecting sorted cells with the corresponding siRNA (Fig 5B). Because the E2F family is known to control cell cycle and because of the predicted downstream genes enriched for cell cycle from the GO analysis (Table S2), we decided to test the silencing effect on the proliferation of FGFR4+-sorted cells. We transfected FGFR4+-sorted cells with the corresponding siRNAs and 2 d later passaged and stained the cells with the cell tracker CSFE. Cells were then cultured for 1 wk, and fluorescence was analyzed by flow cytometry. We found the transduction of any of the three siRNAs reduced cell division, but the effect was biggest with E2F1 siRNA (Fig 5C). These results were confirmed by counting the number of cells 1 wk after the siRNA transfection (Figs 5 and

S6C), indicating that the E2F family, especially E2F1, plays a role in regulating the proliferation of hiPSC-MuPCs.

To better understand the possible role of E2F genes in muscle stem cell proliferation, we studied their expression in a scRNA-seq atlas of regenerating mouse skeletal muscle (accession code: GSE143437) (De Micheli et al, 2020). We found that the expression of E2F genes was increased in Pax7+ cells 2 d after injury (Fig S6D), probably to stimulate cell proliferation. Seven days after injury, the percentage of Pax7+ cells expressing E2F genes returned to the pre-injury state.

Finally, we evaluated if E2F silencing could affect myogenic differentiation by inducing myogenic differentiation 2 d after the siRNA transfection. Five days after the differentiation, histochemical analysis revealed no effect by the siRNAs (Fig 5F and G), suggesting that the E2F family plays no role in the myogenic differentiation of hiPSC-MuPCs.

## Discussion

HiPSCs can be differentiated into muscle progenitors with great potential for clinical application (Darabi et al, 2012; Hicks et al, 2018; Al Tanoury et al, 2020; Zhao et al, 2020; Nalbandian et al, 2021). However, hiPSC-MuPCs are a heterogeneous population, and it was unknown which subset of the hiPSC-MuPCs are more suitable for cell transplantation. In this study, by performing scRNA-seq of the hiPSC-MuPC cultures, we separated the heterogeneous population of hiPSC-MuPCs into four types: *noncycling progenitors*, *cycling*, *committed*, and *myocytes*. Notably, in the in vitro culture of hiPSC-MuPCs, we could find all these four populations coexisting under homeostatic conditions. This property resembles the fetal stage, during which a large number of MuPCs proliferate but some others are committed to myogenic differentiation for muscle tissue formation (Xi et al, 2020). In contrast, in adults, the heterogeneity can be only found during the regeneration process (De Micheli et al, 2020).

Moreover, by using our previously reported surface marker, FGFR4 (Nalbandian et al, 2021), which we found to be expressed by *noncycling progenitors* and *cycling* cells, and the newly identified marker CD36, which is expressed by *myocytes* and some of the *committed* cells, we could separate two populations of hiPSC-MuPCs. FGFR4+ cells showed a higher regenerative capacity and presented stem cell–like characteristics, such as higher proliferation capacity and smaller size, compared with CD36$^+$ cells. These findings are consistent with differences between primary fetal MuPCs and cultured myoblasts (Hicks et al, 2018).

The transcriptional control of gene expressions during myogenic commitment through fetal development and in hiPSC-MuPCs is still not fully understood. By analyzing the TF–binding sites in the promoter regions of DEGs up-regulated in each myogenic cluster, we could identify TFs whose expression is correlated with predicted downstream gene expressions. In particular, E2F family genes were highly expressed by *cycling* cells. We found that several promoter regions of up-regulated genes in *cycling* cells possess binding motifs for the E2F family. Functional studies confirmed that the E2F family, especially E2F1, controls cell proliferation in hiPSC-MuPCs.

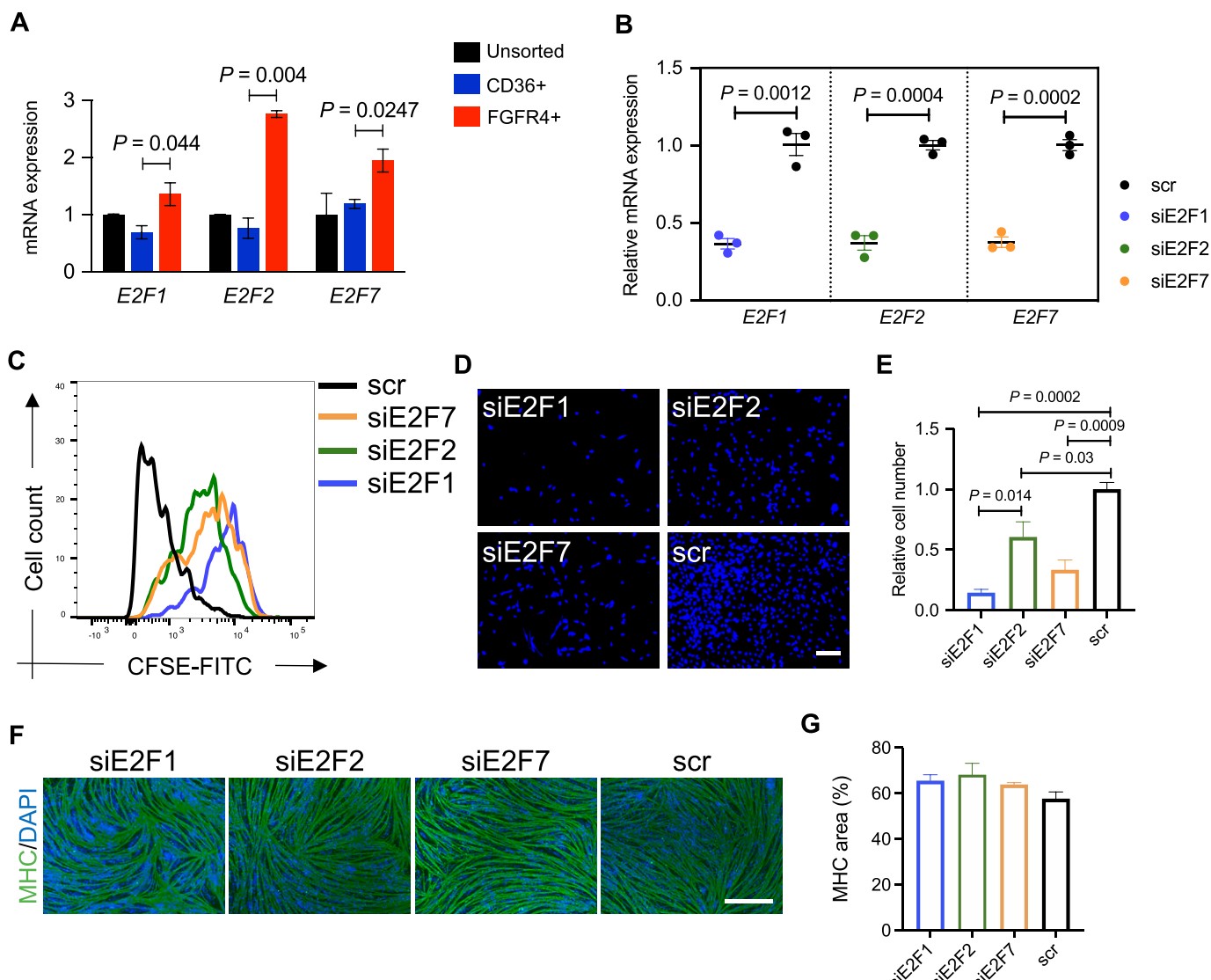

**Figure 5. Knockdown of *E2F1*, *E2F2*, and *E2F7* in FGFR4+ hiPSC-derived muscle progenitor cells.**
**(A)** Expressions of *E2F1*, *E2F2*, and *E2F7* in FGFR4+-sorted hiPSC-MuPCs. The expressions were normalized with *GAPDH*. **(B)** Expressions of *E2F1*, *E2F2*, and *E2F7* in FGFR4+-sorted hiPSC-MuPCs transfected with siRNA for *E2F1* (siE2F1), *E2F2* (siE2F2), *E2F7* (siE2F7), or scramble siRNA (scr). The expressions were normalized to *GAPDH*. **(C)** Cell division tracking by CSFE staining. FGFR4+-sorted hiPSC-MuPCs transfected with the corresponding siRNAs were stained for CSFE and cultured for 1 wk. Then, the CSFE intensity was detected by flow cytometry. **(D)** Representative images of DAPI stained FGFR4+-sorted hiPSC-MuPCs 1 wk after transfection with the corresponding siRNAs. Scale bar, 200 µm. **(E)** Quantification of (D). **(F)** Representative images of differentiated FGFR4+-sorted hiPSC-MuPCs transfected with the corresponding siRNAs. Two days after the transfection, the medium was changed to differentiation medium (2% HS), and the cells were cultured for 5 d until myogenic differentiation. Myosin heavy chain (MHC, green), DAPI (blue). Scale bar, 200 µm. **(G)** Quantification of (F). Error bars in (A, B), (E), and (G) represent the mean ± SEM of three independent experiments (n = 3).

Previous studies showed similar results with other myogenic progenitors. For instance, E2f family genes are reported to be up-regulated at the early stages of muscle regeneration in a cardiotoxin injury mice model (Yan et al, 2003; De Micheli et al, 2020). Moreover, when ablating *E2f1* in mice, skeletal muscle regeneration was severely impaired but not when ablating *E2f2* (Yan et al, 2003). In addition, we identified TFAP4 as a TF that may play an important role in muscle development. *TFAP4* was mainly expressed by *noncycling progenitors* and *cycling* cells, but its binding sites were enriched in the promoter regions of genes up-regulated in *committed* and *myocytes* cells, suggesting a possible role in the transition to myogenic commitment. Although TFAP4 has been

reported in cancer cells to mediate cell fate decisions by diverse mechanisms including the PI3K/Akt pathway (Huang et al, 2019; Wong et al, 2021), to our knowledge, TFAP4 has not previously been studied in muscle progenitor cells. Future studies should address the possible function of TFAP4 in the myogenic process.

Using a pseudo-time analysis, we inferred that *cycling* cells are the progenitors of *committed* and *myocytes* cells. Interestingly, *cycling* cells were situated earlier than *noncycling progenitor* cells in the pseudo-time analysis, suggesting the possibility that *cycling* cells may give rise to *noncycling progenitor* cells too. Considering the development of skeletal muscle, *cycling* cells could be progenitors for developing muscle fibers by differentiating to the

myogenic branch (*committed* and *myocytes* cells) and the progenitors for future adult skeletal muscle stem cells (satellite cells) by differentiating to the noncycling *progenitor's* branch (*noncycling progenitors* cells). Yet, whether *noncycling progenitors* cells really exit the cell cycle and become adult quiescent cells requires future study, as does the possibility of a dynamic transition between *noncycling progenitors* and *cycling* cells. Future studies should evaluate hiPSC-MuPCs at different time points using a multi-omics analysis and cell tracking to bring light to these questions.

In summary, this study established a single-cell atlas of hiPSC-MuPCs and identified four different myogenic populations. We further identified FGFR4+ cells as representative of a subset of hiPSC-MuPCs that have higher in vivo regeneration potential. Finally, by a bioinformatic analysis, we identified the E2F TF family as key players in hiPSC-MuPC proliferation. These results have implications for the quality of cells used in cell therapies toward regenerative medicine and for the understanding of hiPSC-MuPC biology.

# Materials and Methods

### Mouse models

For the animal experiments, immunosuppressed dystrophic male NOG-mdx mice were used. The animals were 6 to 8 wk old at the time of the transplantation.

### HiPSC lines

The hiPSC lines used in this study were a DMD-corrected cell line (Li et al, 2015) and Ff-WJs516 (abbreviated as S516 in this manuscript) and 414C2. The DMD-corrected cell (Li et al, 2015) line was generated by knocking-in exon 44 into a DMD patient-derived hiPS cell line lacking exon 44 (which was generated from the dermal fibroblast using episomal vectors). S516 is homozygous for the most frequent HLA haplotype in Japan and was established from cord blood cells using an episomal vector system as previously described (Okita et al, 2011). It was generated under written consent with the approval by the Kyoto University Graduate School and Faculty of Medicine, Ethics Committee (approval numbers #E1762, #G567and #Rinsho71). 414C2 was established from dermal fibroblasts using the same episomal vector system (Okita et al, 2011). It was also used to establish the *PAX*-Venus reporter line (Nalbandian et al, 2021).

### In vitro stepwise differentiation protocol

The myogenic induction of hiPSCs was performed using a previously described transgene-free protocol (Zhao et al, 2020). Briefly, hiPSCs were seeded in Matrigel-coated wells of a six-well plate and cultured with StemFit (AK02N; Ajinomoto) medium (1 × 10⁴ cells/well). At day 3, the medium was changed to CDMi medium supplemented with CHIR99021 (CHIR, Axon MedChem; Tocris) and SB431542 (SB; Sigma-Aldrich). CDMi medium is composed of IMDM (12440053; Invitrogen) and F12 (1X) Nutrient Mixture (Ham) (11765054; Invitrogen) at the ratio 1:1 supplemented with 1% BSA (Sigma-Aldrich),

1% Penicillin Streptomycin Mixed Solution (Nacalai), 1% CD Lipid Concentrate (Invitrogen), 1% Insulin-Transferrin Selenium (Invitrogen), and 450 $\mu$M 1-Thioglycerol (Sigma-Aldrich). One week later, the cells were dissociated with Accutase and passed to a Matrigel-coated dish with CDMi medium supplemented with SB and CHIR (8 × 10⁵ cells/well). One week later, the cells were dissociated with Accutase and passaged to Matrigel-coated wells in a six-well plate with CDMi medium (8 × 10⁵ cells/well). Three days later, the medium was switched to SFO3 medium (SF-O3; Sanko Junyaku) supplemented with IGF-1, bFGF, and HGF. At day 35 of the differentiation, the medium was switched to DMEM (11960069; Invitrogen) supplemented with 0.5% Penicillin-Streptomycin (26253-84; Nacalai), 2 mM L-glutamine (16948-04; Nacalai), 0.1 mM 2-ME, 2% Horse Serum (HS; Sigma-Aldrich), 5 $\mu$M SB, and 10 ng/ml IGF-1. This medium was replaced with fresh medium of the same composition three times per week until the cells were used for the subsequent experiments.

### Single-cell RNA sequencing

S516 was used for scRNA-seq. At day 80 of the myogenic differentiation, single cells were acquired upon incubation for 1 h with DMEM medium with Collagenase G (500 $\mu$g/ml) and H (500 $\mu$g/ml) (Meiji), followed by 10 min with Accutase (Nacalai) at 37°C. Then the cells were carefully detached by pipetting, filtered with a 50-nm mesh and washed twice with 1% BSA in HBSS (Gibco). The cells were resuspended in 1% BSA in HBSS to reach a concentration of 1,000 cells/$\mu$l. The cDNA library was prepared using the Next GEM Single Cell 3′ Gel Bead Kit v3.1 (1000129), Chromium Next GEM Chip G Single Cell Kit v3 (PN-1000127), Next GEM Single Cell 3′ GME Kit v3.1 (1000130), Next GEM Single Cell 3′ Library Kit v3.1 (1000158), and i7 Multiplex Kit (PN-120262) (10x Genomics) according to the 10x Genomics instructions. Then the cDNA library was run on an Illumina NextSeq 500 and HiSeq 4000.

### Analysis of single-cell RNA sequencing data

The sequenced reads were demultiplexed, mapped, and quantified into UMI-filtered counts using Cell Ranger pipelines (v.4.0.0; 10x Genomics) with the hg38 human reference genome. Scrublet (v.0.2.3) was used to exclude doublet cells (Scrublet score ≥ mean + 1SD) from further analysis. The raw counts data of the filtered cells were further analyzed with the Seurat package (v.4.0.1) (Butler et al, 2018). For further quality control of the extracted gene-cell matrices, we filtered the cells with a low threshold = 3,000 for the number of detected genes per cell (nFeature_RNA), a low threshold = 10,000 and high threshold = 100,000 for the number of UMIs per cell (nCount_RNA), and a high threshold = 10 percent for mitochondrial genes (percent.mito). As a result, 2,406 cells and 2,907 cells for experiment1 and experiment2, respectively, were used. Raw counts were normalized using the LogNormalize method and scaled using the ScaleData function in the Seurat package. UMAP analysis and clustering were performed using the Seurat RunUMAP function with default parameters (except dims = 1:50) and "FindClusters" function with the resolution set to 0.3. Monocle2 (version 0.2.0) (Qiu et al, 2017) was used for the trajectory analysis. For the analysis of the cell cycle, the method reported by Kowalczyk and colleagues was used (Kowalczyk et al, 2015). Based on the gene expression of distinctive

markers for cell cycle, each cell was given a score for the G2/M and S phase. Cells that did not express G2/M and/or S phase markers were identified to be in G1 phase.

DEGs were defined by *P*-value < 0.05 and twofold change. Pathway enrichment analysis and TF-binding site motif analysis were performed using the online tool Enrich: https://maayanlab.cloud/Enrichr/ (Xie et al, 2021).

### Cell sorting

Cells were dissociated by treating them with a mixture of Collagenase G and H for 5 min, followed by 7 min of incubation with TrypLE (Thermo Fisher Scientific). Then the cells were carefully dissociated by pipetting and washed with HBSS. After two rounds of HBSS washing, the cells were incubated for 20 min on ice with conjugated antibodies for FGFR4-PE and/or CD36. After the cells were incubated with antibodies, they were washed twice with HBSS, and cells pellets were resuspended with HBSS with Hoechst and sorted with an ARIA 2 flow cytometer.

### Cell transplantation

A total of 50,000 FGFR4+ or CD36⁺ hiPSC-MuPCs sorted cells were suspended in 50 $\mu$l of DMEM and injected into the cryo-injured TA muscle of NOG-mdx mice. The cryo-injury consisted of applying pressure with a forceps chilled with liquid nitrogen to an exposed muscle for three bouts 10 s long each. Four weeks later, the mice were sacrificed, and histological analysis was performed.

### In vitro differentiation

After sorting, CD36⁺ or FGFR4+ hiPSC-MuPCs were plated onto a laminin 511-precoated 96-well dish (10,000 cells/well) and cultured with StemFit for 5 d. At day 5, the medium was changed to differentiation medium (2% HS). The cells were cultured for five more days for myogenic differentiation.

### Real time RT-qPCR

mRNA was obtained from the cells using the ReliaPrep RNA Cell Miniprep System (Z6012; Promega). After mRNA extraction, cDNA was synthesized using the ReverTra Ace qPCR RT Kit (FSQ-101; TOYOBO). Real time RT-qPCR was performed using a One Step thermal cycler (Applied Biosystems) with the SYBR Green System (Applied Biosystems). Primers used for the real time RT-qPCR are listed in Table S3.

### Immunostaining

For the histochemical analysis, samples were first fixed with 2% PFA for 10 min, then washed two times with PBS, and blocked with Blocking One for 1 h. After blocking, the samples were incubated with the first antibody for 1 h at room temperature. After the first antibody incubation, samples were washed three times with PBS-T and incubated for 1 h with secondary antibodies and DAPI at room temperature. Later, the samples were washed one time with PBS-T

and two times with PBS. Finally, the stained samples were observed by microscopy. The antibodies used are listed in Table S4.

### siRNA transfection

For the knockdown experiments, cells were seeded for 24 h after sorting and transfected with the corresponding siRNA (20 nM) (Cat. no. s4405, Cat. no. s4408, Cat. no. 44665, Cat. no. 4390843) using the Lipofectamine RNAiMAX reagent (Thermo Fisher Scientific). Twenty-four hours after the transfection, the medium was changed, and the cells were cultured for one more day and prepared for the subsequent experiments.

### CFSE staining

Two days after the siRNA transfection, the hiPSC-MuPCs were passaged. During the passaging and while in suspension, the cells were incubated with CFSE working solution (423801; BioLegend) for 20 min at 37°C and protected from light. Then the cells were seeded in a laminin 511-coated 24-well plate. One week later, the fluorescence was analyzed by flow cytometry.

### Cell cycle analysis

After dissociating the hiPSC-MuPCs and staining for FGFR4, the cells were stained with Cell Cycle Solution Blue (Doijindo Molecular Technologies, Inc. C549) for 15 min at 37°C. After that, cells were analyzed with the ARIA 2 flow cytometer.

### Quantification and statistical analysis

All data analyzed in this article are from at least three independent experiments. All statistical analyses were performed using GraphPad Prism version 8.4.1 for Mac OS X (GraphPad Software). For a comparison between two groups, a *t* test was performed. For a comparison of three or more groups, an ANOVA (analysis of variance) with Tukey's range test for multiple comparisons was performed. Significant differences were considered when the *P*-value was <0.05. Flow cytometry analysis was performed by using FlowJo software.

## Data Availability

The accession number for the single-cell RNA sequencing reported in this paper has been deposited in the Gene Expression Omnibus (GEO) database GSE199467. Correspondence and requests for materials should be addressed to H Sakurai.

## Supplementary Information

# Acknowledgments

We would like to thank to Rukia Ikeda and Mikiko Fukuda for their contribution in the establishment of the hiPSC reporter line; to Kanae Mitsunaga for her constant help with the FACS experiments; and to Peter Karagiannis for proofreading the manuscript. This work was mainly supported by a grant from The Core Center for iPS Cell Research (JP21bm0104001), which is a program in the Research Center Network for Realization of Regenerative Medicine, Japan Agency for Medical Research and Development (AMED; to H Sakurai).

## Author Contributions

M Nalbandian: conceptualization, data curation, formal analysis, investigation, visualization, methodology, and writing—original draft, review, and editing.
M Zhao: investigation, methodology, and writing—review and editing.
H Kato: conceptualization, investigation, and methodology.
T Jonouchi: investigation and methodology.
M Nakajima-Koyama: investigation and methodology.
T Yamamoto: resources, data curation, formal analysis, investigation, visualization, methodology, and writing—review and editing.
H Sakurai: conceptualization, resources, supervision, funding acquisition, project administration, and writing—review and editing.

## Conflict of Interest Statement

The authors declare that they have no conflict of interest.

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
