## [Reviewer comments · Life Science Alliance]

Life Science Alliance

Single-cell RNAseq reveals heterogeneity in hiPSC-MuPCs and E2F as a key regulator of proliferation

Minas Nalbandian, Mingming Zhao, Hiroki Kato, Tatsuya Jonouchi, May Nakajima- Koyama, Takuya Yamamoto, and Hidetoshi Sakurai

DOI: <https://doi.org/10.26508/lsa.202101312>

Corresponding author(s): Hidetoshi Sakurai, Center for iPS Cell Research and Application, Kyoto University and Minas Nalbandian, Center for iPS Cell Research and Application, Kyoto University

Review Timeline:

Submission Date:	2021-11-22
Editorial Decision:	2022-01-17
Revision Received:	2022-02-16
Editorial Decision:	2022-03-14
Revision Received:	2022-03-28
Accepted:	2022-03-30

Scientific Editor: Novella Guidi

Transaction Report:

January 17, 2022

Re: Life Science Alliance manuscript #LSA-2021-01312-T

Dr. Hidetoshi Sakurai
Center for iPS Cell Research and Application, Kyoto University
Department of Clinical Application
53, Shogoin Kawahara-cho
Sakyo-ku
Kyoto, Kyoto 606-8507
Japan

Dear Dr. Sakurai,

Thank you for submitting your manuscript entitled "Single-cell RNAseq reveals heterogeneity in hiPSC-MuPCs and E2F as a key regulator of proliferation" to Life Science Alliance. The manuscript was assessed by expert reviewers, whose comments are appended to this letter. We, thus, encourage you to submit a revised version of the manuscript back to LSA that responds to all of the reviewers' points.

Thank you for this interesting contribution to Life Science Alliance. We are looking forward to receiving your revised manuscript.

Sincerely,

B. MANUSCRIPT ORGANIZATION AND FORMATTING:

Reviewer #2 (Comments to the Authors (Required)):

In this manuscript the authors have performed single cell RNA sequencing of hiPSC-MuPC cultures that revealed four distinct clusters among which FGFR4-positive dormant and cycling cells show higher regenerative capacity than CD36-positive myoblast and committed cells. Moreover, the data revealed an essential role of E2F transcription factors as regulators of hiPSC-MuPC proliferation.

Overall, this study is well performed and in general I am in favor of publication in LSA. Although the major findings and conclusions lack substantial novelty - i.e. the higher regenerative capacity of muscle stem cells vs committed myoblasts has been largely demonstrated by several studies; the regulation of cell proliferation by E2F family of transcription factors is well known - the data generated expand the knowledge on a topic of current interest and translational importance.

The authors have an opportunity to improve this manuscript tremendously by providing insights into the role of E2F family members in the regulation of the regenerative capacity of the four cellular clusters of hiPSC-MuPC identified here. For instance, the authors might test the effect of RNAi-mediated downregulation of individual E2F family member of specific interest in cellular cluster prior to transplanting cells in vivo, and determine the effect as compared to control scramble RNAi.

Other minor points

The efficacy of RNAi-mediated knockdown of each E2F family member needs to be validated by monitoring protein levels or at least RNA levels.

The reference "Zhao, M., Tazumi, A., Takayama, S., Takenaka-Ninagawa, N., Nalbandian, M., Nagai, M., Nakamura, Y., Nakasa, M., Watanabe, A., Ikeya, M., et al. (2020). Induced Fetal Human Muscle Stem Cells with High Therapeutic Potential in a Mouse Muscular Dystrophy Model. *Stem Cell Reports*. - is missing details.

Reviewer #3 (Comments to the Authors (Required)):

The manuscript by Nalbandian et al. is a follow up study using a protocol previously developed by this group to differentiate hiPSCs into fetal-like myogenic cells that harbor muscle stem cell attributes and can engraft in vivo (Zhao et al 2020). In another recently published work by this group (Nalbandian et al. *Stem Cell Reports* 2021), the authors characterized the cell types comprising the iPSC-differentiated cultures using a double genetic reporter (Pax7 / Myf5) and RNA-Seq. Further, they identified FGFR4 as a surface marker that enables isolation of muscle stem cells from the heterogenous hiPSC-differentiated cultures.

In the current study the authors embarked on further characterization of the differentiation process using single cell RNA-Seq. Via this approach, they characterize the various cell types comprising the differentiated cultures (neural, mesenchymal and myogenic), and further define four stages indicative of myogenesis specifically in the muscle cell population (Myoblast, Committed, Cycling, and Dormant). The authors corroborate the presence of FGFR4 predominantly in the myogenic stem cell fraction and identify CD36 as a surface marker associated with skeletal muscle differentiation. They demonstrate that FGFR4+ cells harbor an augmented capacity to differentiate in vitro and in vivo into multinucleated skeletal muscle cells in comparison to CD36+ cells. Finally, the authors identify the cell cycle and transcription factor E2F family as key for the proliferation, however not differentiation of hiPSC-derived myogenic cells.

This work represent a logical step forward in regards to the characterization of iPSC-derived myogenic cells, which is of interest to the field as such cells are deemed suitable for potential cell therapy and disease modeling. The work corroborates other recent studies via scRNA-Seq that show that hiPSCs-derived myogenic cells represent a fetal rather than adult cell stage, and the data presented by the authors supports their major conclusions. I have several suggestions and queries before fully endorsing this manuscript for publication:

1) The authors base their identification of non-myogenic cell types in the heterogenous hiPSC-derived cultures solely on one

marker, i.e. sox2 for neural, PDGFR4A for mesenchymal cells. It will be helpful if additional markers associated with each cell type are provided.

II) The definition of "dormant" is a bit misleading given the cells might fluctuate between cycling and non-cycling cell states, and not enter a quiescence phase that is associated with adult muscle satellite cells. As such, it may be more accurate to denote the dormant cells as "non-cycling precursors". Further, can the authors provide a scatter plot of differentially expressed genes between the "cycling" and non-cycling / "dormant" cell populations? It will be of interest if various satellite cell markers are differently expressed between the "dormant" cell population and the "cycling" one.

III) Satellite cell-derived myoblasts represent a proliferative cell population that is still not fully committed into a myogenic differentiation program. Given the cells termed "myoblasts" by the authors downregulate cell cycle regulators in addition to Pax7 and Myf5 while upregulating eMHC, ACTA1, MYMX and MYMK they may in fact represent a population of "myocytes" or "early differentiated muscle cells" and not myoblasts. The author should consider changing their definition of this cell population to another more accurate term.

IV) The authors indicate that FGFR4+ cells manifest enhanced myogenic differentiation in comparison to CD36+ cells via quantification of DAPI+ cells associated with MHC positive cells. However, this can be attributed to a higher number of fibers due to the increased proliferation capacity of FGFR4+ vs. CD36+ cells. As such, it will be helpful if the authors could also quantify the number of DAPI+ cells per individual myofiber for several MHC+ multinucleated fibers generated from FACS-sorted FGFR4+ or CD36+ cells.

V) Can the authors report the expression of Pax3 in their single cell RNA-Seq data? Expression of this marker may indicate presence of a more immature cell population in the culture.

Minor comments:

i. Please label the X axis in Fig. 4G or add a legend.

ii. Please provide 1-2 additional markers for proliferation in Figure 2.

iii. Please explain in more details how DEGs were defined using the scRNA-Seq.

iv. "DMD corrected" is mentioned without reference to prior work, which seems a bit out of context if one is not familiar with the author's previous works.

v. Could the author clarify the difference between Fig 4J, 4K and 4L and S4F, S4G and S4H?

iv. Further discussion revolving around the newly identified marker CD36 will be helpful.

Point-by point response**Response to Reviewer #2 comments:**

We thank Reviewer 2 for taking the time to read our manuscript. We appreciate his/her comments and corrections, which we tried to incorporate in the new version of the manuscript.

The authors have an opportunity to improve this manuscript tremendously by providing insights into the role of E2F family members in the regulation of the regenerative capacity of the four cellular clusters of hiPSC-MuPC identified here. For instance, the authors might test the effect of RNAi-mediated downregulation of individual E2F family member of specific interest in cellular cluster prior to transplanting cells in vivo, and determine the effect as compared to control scramble RNAi.

We appreciate the reviewer's comment and agree that that experiment would provide strong evidence into the E2F role in iPSC-MuPCs towards cell therapy. However, unfortunately we cannot do this experiment because of technical limitations. After cell sorting, the cells need to be cultured for at least 24 hours to be transfected (when transfected immediately after sorting, the cells die). On top of that, our lab has shown that when hiPSC-MuPCs are cultured after sorting, the cell quickly loses its in vivo regeneration potential.

Other minor points

The efficacy of RNAi-mediated knockdown of each E2F family member needs to be validated by monitoring protein levels or at least RNA levels.

We check mRNA levels 48 hours after transfection and it is reported in figure 4B, which is the same time that the medium was switched to differentiation medium. Therefore, we assume that the silencing was present at least during the first days of differentiation. According to the Thermo fisher website, the silencing is present for at least 5 days post-transfection (<https://www.thermofisher.com/us/en/home/references/ambion-tech-support/rnai-sirna/tech-notes/duration-of-sirna-induced-silencing.html>). Besides we did not check the silencing during the differentiation, our data suggest that differentiation was not affected by silenced E2F1,2 and 7 expressions at the beginning of the differentiation.

The reference "Zhao, M., Tazumi, A., Takayama, S., Takenaka-Ninagawa, N., Nalbandian, M., Nagai, M., Nakamura, Y., Nakasa, M., Watanabe, A., Ikeya, M., et al. (2020). Induced Fetal Human Muscle Stem Cells with High Therapeutic Potential in a Mouse Muscular Dystrophy Model. Stem Cell Reports. - is missing details.

We completed the reference details in the new version.

Response to Reviewer #3 comments:

We thank Reviewer 3 for taking the time to read our manuscript. We appreciate his/her comments and corrections, which we tried to incorporate in the new version of the manuscript.

I) The authors base their identification of non-myogenic cell types in the heterogeneous hiPSC-derived cultures solely on one marker, i.e. sox2 for neural, PDGFR4A for mesenchymal cells. It will be helpful if additional markers associated with each cell type are provided.

In the new version, we included more markers for Mesenchymal cells (PDGFRB, ENG, COL6A3, and COL6A2) as well as for Neuronal cells (PAX6, SOX1, SOX3, and SOX5). The plots are included in Figure S1C.

II) The definition of "dormant" is a bit misleading given the cells might fluctuate between cycling and non-cycling cell states, and not enter a quiescence phase that is associated with adult muscle satellite cells. As such, it may be more accurate to denote the dormant cells as "non-cycling precursors".

We changed the denomination of "Dormant" to "non-cycling".

Further, can the authors provide a scatter plot of differentially expressed genes between the "cycling" and non-cycling / "dormant" cell populations? It will be of interest if various satellite cell markers are differently expressed between the "dormant" cell population and the "cycling" one.

We included a Volcano plot showing DEGs between "non-cycling" and "cycling" cells, and GO analysis. Please check Figure S2D and Table S1. We could not identify any satellite cell marker of myogenic cell population-related gene on the DEGs between cycling and non-cycling cells.

III) Satellite cell-derived myoblasts represent a proliferative cell population that is still not fully committed into a myogenic differentiation program. Given the cells termed "myoblasts" by the authors downregulate cell cycle regulators in addition to Pax7 and Myf5 while upregulating eMHC, ACTA1, MYMX and MYMK they may in fact represent a population of "myocytes" or "early differentiated muscle cells" and not myoblasts. The author should consider changing their definition of this cell population to another more accurate term.

We changed the denomination of "Myoblast" to "Myocytes".

IV) The authors indicate that FGFR4+ cells manifest enhanced myogenic differentiation in comparison to CD36+ cells via quantification of DAPI+ cells associated with MHC positive cells. However, this can be attributed to a higher number of fibers due to the increased proliferation capacity of FGFR4+ vs. CD36+ cells. As such, it will be helpful if the authors could also quantify the number of DAPI+ cells per individual myofiber for several MHC+ multinucleated fibers generated from FACS-sorted FGFR4+ or CD36+ cells.

We included the average nuclei per myofiber (Figure 4M, and S5I). We found that the FGFR+ cells increased the nuclei number per myofiber.

V) Can the authors report the expression of Pax3 in their single cell RNA-Seq data? Expression of this marker may indicate presence of a more immature cell population in the culture.

There is non-significant PAX3 expression in the myogenic cells, indicating that possible, most of the cells are not at the embryonic stage.

Minor comments:

i. Please label the X axis in Fig. 4G or add a legend.

We added labels to the new version.

ii. Please provide 1-2 additional markers for proliferation in Figure 2.

We included 4 new markers in figure S2B.

iii. Please explain in more details how DEGs were defined using the scRNA-Seq.

The DEGs we defined as $P > 0.05$ - and 2-fold change. We include this description in the method sections.

iv. "DMD corrected" is mentioned without reference to prior work, which seems a bit out of context if one is not familiar with the author's previous works.

We included a description when introducing them for the first time on the manuscript.

v. Could the author clarify the difference between Fig 4J, 4K and 4L and S4F, S4G and S4H?

The experiments were performed with different cell lines. We clarified that on the new version.

iv. Further discussion revolving around the newly identified marker CD36 will be helpful.

We included a sentence in the result section describing the reported role of CD36 in myoblasts fusion.

March 14, 2022

RE: Life Science Alliance Manuscript #LSA-2021-01312-TR

Dr. Hidetoshi Sakurai
Center for iPS Cell Research and Application, Kyoto University
Department of Clinical Application
53, Shogoin Kawahara-cho
Sakyo-ku
Kyoto, Kyoto 606-8507
Japan

Dear Dr. Sakurai,

Thank you for submitting your revised manuscript entitled "Single-cell RNAseq reveals heterogeneity in hiPSC-MuPCs and E2F as a key regulator of proliferation". We would be happy to publish your paper in Life Science Alliance pending final revisions necessary to meet our formatting guidelines.

- Please upload all figure files as individual ones, including the supplementary figure files
- please add a Running Title to our system
- please add ORCID ID for secondary corresponding author-they should have received instructions on how to do so
- please add the Twitter handle of your host institute/organization as well as your own or/and one of the authors in our system
- Please provide the accession number for the single-cell RNA-sequencing in the data availability section

A. FINAL FILES:

B. MANUSCRIPT ORGANIZATION AND FORMATTING:

Sincerely,

Reviewer #2 (Comments to the Authors (Required)):

Although the authors have not addressed the experimental points I suggested, I agree that the data presented as they stand provide an information of interest that should be made available to the scientific community.

March 30, 2022

RE: Life Science Alliance Manuscript #LSA-2021-01312-TRR

Dr. Hidetoshi Sakurai
Center for iPS Cell Research and Application, Kyoto University
Department of Clinical Application
53, Shogoin Kawahara-cho
Sakyo-ku
Kyoto, Kyoto 606-8507
Japan

Dear Dr. Sakurai,

Thank you for submitting your Research Article entitled "Single-cell RNAseq reveals heterogeneity in hiPSC-MuPCs and E2F as a key regulator of proliferation". It is a pleasure to let you know that your manuscript is now accepted for publication in Life Science Alliance. Congratulations on this interesting work.

DISTRIBUTION OF MATERIALS:

Again, congratulations on a very nice paper. I hope you found the review process to be constructive and are pleased with how the manuscript was handled editorially. We look forward to future exciting submissions from your lab.

Sincerely,
